# Scaling Laws and Compute-Optimal Training Beyond Fixed Training Durations

**Alexander Hägele**[1]*    **Elie Bakouch**[2]    **Atli Kosson**[1]    **Loubna Ben Allal**[2]
**Leandro Von Werra**[2]    **Martin Jaggi**[1]
[1]EPFL    [2]Hugging Face
*`alexander.hagele@epfl.ch`

## Abstract

Scale has become a main ingredient in obtaining strong machine learning models. As a result, understanding a model's scaling properties is key to effectively designing both the right training setup as well as future generations of architectures. In this work, we argue that scale and training research has been needlessly complex due to reliance on the cosine schedule, which prevents training across different lengths for the same model size. We investigate the training behavior of a direct alternative — constant learning rate and cooldowns — and find that it scales predictably and reliably similar to cosine. Additionally, we show that stochastic weight averaging yields improved performance along the training trajectory, without additional training costs, across different scales. Importantly, with these findings we demonstrate that scaling experiments can be performed with significantly reduced compute and GPU hours by utilizing fewer but reusable training runs. Our code is available at `https://github.com/epfml/schedules-and-scaling/`.

## 1 Introduction

Training large language models is expensive — in time, energy, and compute. Moreover, it requires a complex algorithmic recipe of model architecture and training data to obtain high-quality models. Therefore, the workflow of training large models consists of iterating over small experiments to verify success before extrapolating to larger scales. This is then either done by computing specialized scaling laws (OpenAI, 2023; Bi et al., 2024; Hu et al., 2024; Team et al., 2023), relying on established laws (Hoffmann et al., 2022; Anil et al., 2023) or training past compute-optimality to save cost at inference (Touvron et al., 2023a,b).

Despite large advances across data and training recipes, one aspect of large language model (LLM) pretraining has remained surprisingly prevalent: the cosine learning rate schedule (Loshchilov & Hutter, 2016; Radford et al., 2018; Rae et al., 2021). Importantly, the Chinchilla project (Hoffmann et al., 2022) showed that the cosine schedule achieves optimal loss *only* when the cycle length *matches* the training duration, but underestimates the model performance during training. This means that when performing experiments — e.g., for architectural changes or data mixtures — one must train multiple models for different lengths, *from scratch*, to have reliable estimates of the quality of training and the scaling behavior. This is much more expensive than training a suite of models just once. Even more, it is restrictive for the final model for which the training length must be decided in advance.

In this work, our goal is to revisit and question the necessity of the cosine learning rate schedule for large model training. Through a multitude of training runs, we demonstrate how a simple alternative of performing a cooldown after a constant learning rate — which was already suggested in the literature (Zhai et al., 2022) and recently used by released models (Hu et al., 2024; Shen et al., 2024) — matches the performance of cosine. We expand on this and provide and analyze different recipes for the decay form and length, which scale as reliable as cosine, outperforming it for sufficiently long

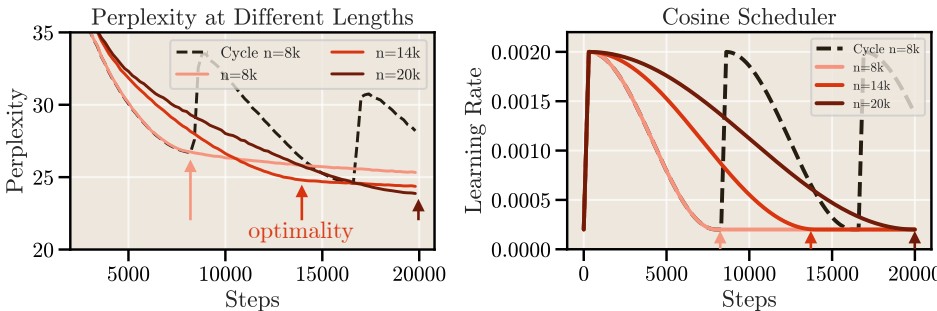

Figure 1: **Revisiting cosine optimality for language models.** We revisit the observation from Chinchilla (Hoffmann et al., 2022) that in order to achieve the best model after a certain training length (tokens), the cosine schedule must match the total duration of training. This comes at the cost of neither being able to stop before or going beyond the cycle — an issue we show how to alleviate in Section 3.

cooldowns. Going beyond, we investigate stochastic weight averaging (Izmailov et al., 2018) and a schedule-free optimizer (Defazio et al., 2024), which give strong (but not optimal) performance at any point during training and can act as a replacement for the learning rate decay, if the performance gap is acceptable and avoiding separate cooldowns is preferred.

These findings suggest that research on training recipes and scaling laws has been needlessly complex due to the need to retrain models from scratch. We demonstrate this empirically by performing a small-scale experiment of scaling laws which only uses a fraction of the compute and GPU hours that were previously needed. Following, we discuss that this makes scaling research more accessible and enables more frequent computation of laws for data mixtures (Bi et al., 2024; Goyal et al., 2024; Aghajanyan et al., 2023) or novel architectures (Gu & Dao, 2023; De et al., 2024).

## 2 Background: Cosine Learning Rate Schedule for LLMs

**Revisiting the optimality of the cosine schedule.** We start our argument by revisiting the use of the cosine schedule in large language model (LLM) training. For any machine learning model, the learning rate value (LR) and schedule both are crucial choices for training. From optimization theory, our understanding is that a slow annealing of the learning rate is essential to find good minima in the loss landscape particularly for deep networks, whereas higher values help exploration (Smith et al., 2017; Loshchilov & Hutter, 2016).

In the context of LLMs, the most commonly used cosine strategy presents a particular trade-off by which the LR reaches its maximum early after the warm-up stage and then gradually decreases, typically to 10% of the maximum LR (see Figure 1, right). Since the seminal works of GPT (Radford et al., 2018, 2019; Brown et al., 2020) and large models like Gopher (Rae et al., 2021), PaLM2 (Anil et al., 2023) or LLaMA (Touvron et al., 2023a,b), cosine has stayed the de-facto standard schedule.

**Experimental setup.** Our first goal is to understand the importance of the length of the schedule for performance of the model. To this end, we implement the common decoder-only transformer (Vaswani et al., 2017) identical to the LLaMa (Touvron et al., 2023a,b) or Noam architecture (Ormazabal et al., 2024). Throughout this paper, we use the AdamW optimizer with weight decay (Kingma & Ba, 2014; Loshchilov & Hutter, 2017) with common LLM training parameters. We train on a subset of SlimPajama (Soboleva et al., 2023) with 6B tokens[1], a cleaned and deduplicated corpus for LLM pretraining, which we split into train and validation sequences and report validation loss (perplexity). We provide all the details in Appendix A.1.

**The pitfalls of cosine.** In the results of Figure 1, we see that the key parameter for the cosine schedule is the length of training: At specific step counts, the best perplexity is always achieved by the cosine schedule that matches the length. This is the main observation of Hoffmann et al. (2022) — in order to achieve the best model for a specific token count, the training duration must be known in advance and match the cosine length. However, this brings particular issues. First, cosine is *suboptimal during training* and underestimates the model's performance for the same token count. At the same time, cosine *strongly complicates continuation of training*. For example, one could easily be mistaken

---

[1] https://huggingface.co/datasets/DKYoon/SlimPajama-6B

to extrapolate a loss curve of a cosine schedule beyond the end of the cycle. The improvement in loss precisely happens because of the LR decay; afterwards, the final learning rate will generally be too low to continue making large progress. In contrast, rewarming leads to spikes of which the training only slowly recovers, similarly reported in the continual learning literature (Ibrahim et al., 2024).

# 3 A Different Route: Constant Learning Rate with Cooldown

Why does the cosine schedule with a single cycle work well for LLM training? Arguably, it provides a good trade-off between a high learning rate and cooling down the model sufficiently, which is expanded proportionally to the training steps.

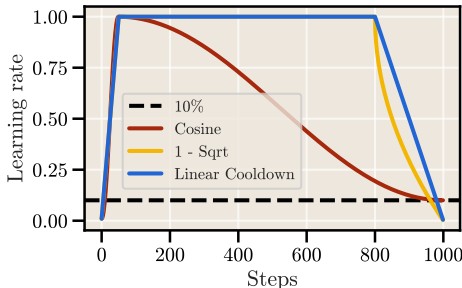

**The alternative — constant + cooldown.** The trade-off of cosine can also be achieved by a different schedule. Here, the learning rate (LR) is kept constant for the majority of the training and only decreases in a short final phase, known as the cooldown (or decay/annealing phase). This schedule has previously been referred to as a trapezoidal (Zhai et al., 2022), and later as the warmup-stable-decay schedule (WSD) (Hu et al., 2024). To avoid overloading of terms (e.g., weight decay), we refer to this approach as constant LR + cooldown.

Figure 2: **Illustration of schedules.** Cosine (red) follows a slow decrease in learning rate, typically to 10% of the maximum for LLMs. The alternative is characterized by an aggressive decrease in learning rate, e.g., via a linear (blue) or square root (yellow) cooldown.

The cooldown phase typically has the LR go to zero, mirroring the warmup phase, which gives rise to an overall trapezoidal shape (see Figure 2). Formally, we can define it as

$$\eta(n) = \begin{cases} \frac{n}{N_{\text{warmup}}} \cdot \eta_{\text{max}} & \text{if } n < N_{\text{warmup}} \\ \eta_{\text{max}} & \text{if } N_{\text{warmup}} < n \leq N - N_{\text{decay}} \\ f(n, N, N_{\text{decay}}) \cdot \eta_{\text{max}} & \text{if } n > N - N_{\text{decay}} \end{cases} \tag{1}$$

with the peak learning rate $\eta_{\text{max}}$, the total steps $N$ with warmup and cooldown steps $N_{\text{warmup}}$ and $N_{\text{decay}}$, and a monotonically decreasing function $f(n, N, N_{\text{decay}})$ that handles the cooldown.

## 3.1 The Advantages

The main advantage of the constant schedule is that it does not require one to specify the number of training steps in advance. This is particularly convenient for large runs, as the cooldown can be initiated at any time to observe model behavior and decide whether to stop. We want to highlight the following in more detail.

**Continual learning.** The constant + cooldown schedule allows for continual learning by default. Here, the natural approach is to use checkpoints before the cooldown to continue training with a high LR; this avoids loss spikes and brittle training when rewarming the learning rate (cf. Figure 1 with cosine). Moreover, rewarming has been reported to hurt performance and introduce forgetting compared to single annealing training (Ibrahim et al., 2024), though careful strategies can alleviate such issues. It remains an interesting question if a single cooldown schedule is absolutely optimal given a total compute budget for LLM training.

**Data mixtures.** During the cooling phase, the data mixture can be changed (Hu et al., 2024) as a form of finetuning; beyond aiming for specific downstream tasks, this allows one to assess the quality of specific mixes, e.g., as recently done for Llama 3 (Dubey et al., 2024). Although we focus on the same mixture, understanding the curriculum aspect of the separate phases is explored in the concurrent work of Blakeney et al. (2024) and remains open for further research.

**Scaling studies.** Since a cooldown can also be initiated retrospectively from a checkpoint, the schedule allows to see the same model at different scales of compute. This enables much cheaper scaling studies than commonly done; we show this in depth in Section 5.

## 3.2 Experimental Comparison

We follow our experimental setup from the previous section (details in A.1) and train a 210M parameter model on SlimPajama with constant LR and the cooldown schedule defined in (1). That is,

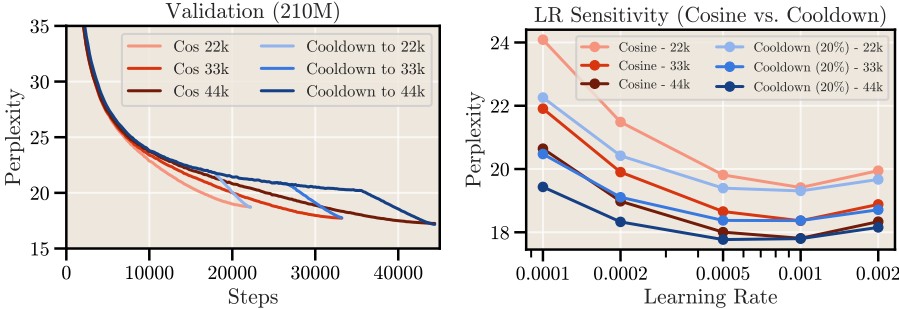

Figure 3: **The difference in loss curves of cosine vs. constant learning rate with cooldown.** The cooldown phase initiates a sharp decrease in loss (left) to match cosine; the training perplexity follows the same behavior (Fig. 15). We find the LR sensitivity (right) to be similar for both schedules, albeit less for cooldown, where the optimum lies slightly below at half of the optimal cosine maximum LR.

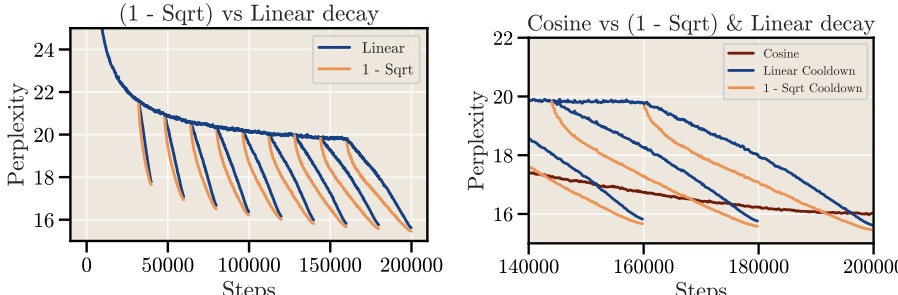

Figure 4: **A different cooldown schedule can improve performance.** Perhaps surprisingly, we find that a different decay phase in the functional form of (1-sqrt) can consistently outperform the standard linear decay, where both are better than the chosen (untuned) cosine for long lengths.

we compare the same length of warmup and training, but replace the cosine decay after warmup with a constant LR and a cooldown for 20% of steps linearly going to zero. We additionally sweep the maximum learning rate for both approaches. In the results shown in Figure 3, perhaps suprisingly, we observe an almost perfect match between the performance of the best cosine and cooldown schedule even for different training durations, all while exhibiting slightly less sensitivity to variations in the LR.

**Different cooldown schedules.** We investigate various functions describing the cooldown shape, including cosine, square, and square root shapes in Appendix B.1. We identify a new function (1-sqrt) that outperforms the linear cooldown. This improvement is maintained in a smaller number of decay steps, different learning rates, and various timestamps (see Appendix B.1). In Figure 4, we train a model for 200K steps (approximately 20B tokens), applying cooldown every 20K steps for 20%. The results indicate that the longer the training duration, the more linear cooldown is outperformed by the (1-sqrt) cooldown, which we define as:

$$f(n, N, N_{\text{decay}}) = \left(1 - \sqrt{\tfrac{n-(N-N_{\text{decay}})}{N_{\text{decay}}}}\right) \qquad \text{(1-sqrt)}$$

> *Takeaway 1:* The constant LR + cooldown schedule offers significant convenience by not requiring the number of training steps to be specified in advance, and provides similar performance compared to a well tuned cosine schedule.

> *Takeaway 2:* We find a cooldown form (1-sqrt) that consistently performs better than linear decay.

**How long do you need to cooldown?** To effectively utilize the schedule, it is essential to determine the optimal number of decay steps. Our study of the relative number of cooldown steps, as shown in Fig. 5, reveals that the benefits of extended cooldown periods plateau at around 20%, which we select for the remaining experiments. Additionally, in Fig. 6 we demonstrate that using only 5% decay with the (1-sqrt) cooldown can nearly match the performance of the cosine schedule on a 20B token run (much beyond Chinchilla optimal). This is practically important, as we ideally keep the cooldown as short as possible but get strong performance.

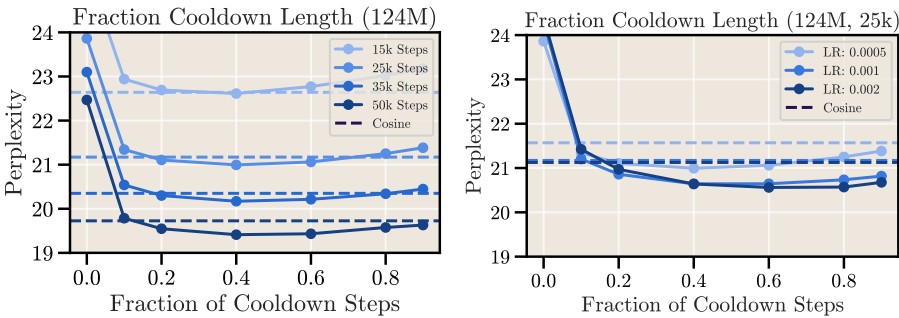

Figure 5: **Longer cooldown helps to achieve lower loss.** We investigate the effect of the cooldown length as a fraction of the total steps for a 124M model. We find that the cooldown surpasses cosine between 10-20% of steps (left), but largely stops improving when done over a majority of training. This also holds when sweeping the LR (right). Additional ablations are provided in Appendix B.1.

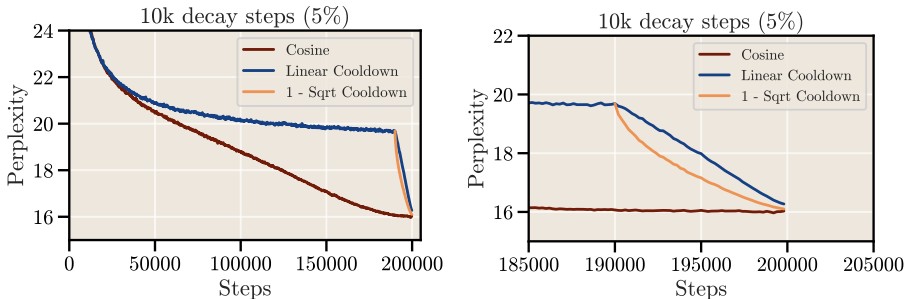

Figure 6: **A long training run suggests that a small number of cooldown steps can match cosine for long training.** From Fig. 5 and Fig. 20, we find that the required duration of cooldown to match the cosine loss *decreases* with longer training; we validate with a long training run (200k steps, same LRs), and find that just 10k cooldown steps almost perfectly match cosine in performance.

> *Takeaway 3:* The constant learning rate with a short cooldown (< 20% of total training steps) can achieve the same final loss as cosine, and only outperforms for a longer fraction of cooldown steps. For long training runs, our results suggest that the cooldown length can be less than 20% to match cosine if it is enough in absolute steps — see Figure 6.

**What happens during the cooldown?** It is remarkable how the sudden drop in loss is consistent for both train and validation loss and aligns closely with the decay in LR (Figure 15). Hu et al. (2024) investigate the cooldown phase and find that the first-order directional derivative diminishes with each step, whereas the curvature of the loss function increases; they attribute this to proximity to a local optimum.

We expand upon these results and aim to understand the optimization landscape around the trajectory of the cooldown. For that, we evaluate the loss along the *straight line trajectory* when moving between a checkpoint before and after cooldown, i.e., a linear interpolation of the weights of the two models. Perhaps surprisingly, we find that the smooth drop in loss also occurs for this interpolation, as visualized in Figure 7. This aligns with the findings of Hu et al. (2024). These results suggest that upon decaying the LR, the model immediately descends into a connected minimum of the loss.

> *Takeaway 4:* The cooldown phase is a smooth transition to a basin in the loss landscape.

**Decaying cosine to less than 10%.** Throughout our experiments, we follow the common heuristic of setting the final LR of cosine to 10% of the maximum. We also provide experiments in which we ablate annealing to zero (or a very small value) in Fig. 22 in Appendix B.1. We find that, perhaps unsurprisingly, the influence of the final LR is non-negligible: decaying to lower values improves performance to match the best (1-sqrt) cooldown. However, when evaluating on downstream benchmarks (see next Section 3.3), we see that annealing to zero can hurt metrics; we posit this comes from too early saturation. Combining these two arguments implies that the maximum and final LR should be set (and ideally swept over) independently.

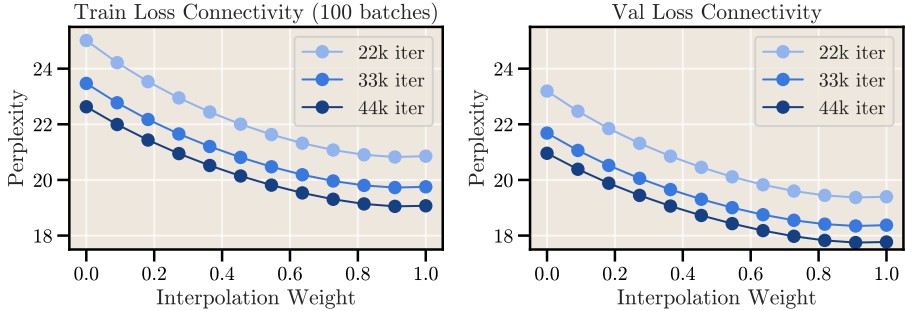

Figure 7: **The smooth drop in loss also occurs when moving linearly in weight space between checkpoints before and after the cooldown.** This suggests that in the cooldown phase, the model directly moves within a connected basin in the loss landscape.

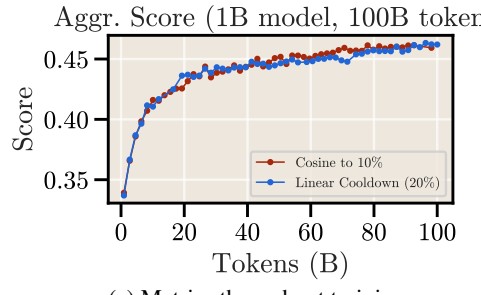

| | Aggregated Score |
|---|---|
| **Cosine to 10%** | 46.26 |
| **Cosine to 0** | 45.88 |
| **Linear (20%)** | 46.20 |
| **(1-Sqrt) (20%)** | 46.23 |

(a) Metrics throughout training.      (b) Final scores (100B tokens).

Figure 8: **The performances of both schedules also match for downstream tasks.** We run a realistic setting of a 1B model trained for 100B tokens of FineWeb and establish matching performance of both schedules for common LLM benchmarks. Interestingly, there is a similar boost in performance with the cooldown. Detailed numbers over the course of training and 460B token runs are in Appx. B.5.

## 3.3 Scaling Up: 1B and 8B Models

We expand our ablations to realistic scenarios of 1B and 8B parameter models, with an additional focus on *model evaluations*. Our goal in this section is therefore twofold: first, we validate the dynamics of the schedule at practical scale; second, we establish the connection between the pure loss values and downstream benchmarks. Although tasks often scale reliably with loss (Du et al., 2024; Gadre et al., 2024), this connection is crucial, as downstream abilities are ultimately the main metric of interest.

**1B with downstream evals.** We train a 1B parameter model with cosine and a cooldown schedule with the high-quality FineWeb corpus (Penedo et al., 2024). Following DeepSeek scaling laws (Bi et al., 2024), we set an approximately optimal batch size and LR of 1.8M and 8e-4 for 100B and 460B token runs; evaluation is done throughout training on the most common benchmarks like MMLU (Hendrycks et al., 2021). The setup is described in detail in Appx. A.2.

With the aggregated results in Fig. 8, we find a clear agreement of the downstream performance for both schedules. Interestingly, there is an uptick for some evaluations, similar to the drop in loss, with the start of the LR cooldown; see Fig. 29. However, not all benchmarks follow this trend, which requires further research. The results for the much longer 460B run, where performance also matches, and detailed benchmark numbers are provided in Appx. B.5.

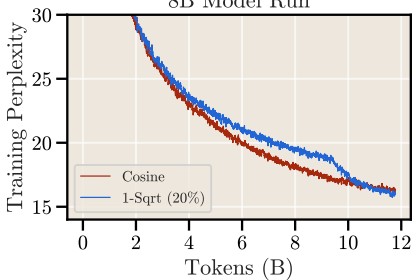

Figure 9: **Results at 8B scale.** We validate the behavior with a much larger model (architecture of Llama 3) for a single short run (12B tokens of FineWeb-Edu), where the cooldown matches the cosine schedule again.

**8B.** While a full 8B training run is beyond our limits, we provide a first investigation in Fig. 9: In line with all our experiments, we see matching loss values and, promisingly, no instability at such a large scale, though the two runs are clearly short (20k steps). Moreover, instabilities arising from a high LR for longer training can be alleviated by methods like QK norm (Wortsman et al., 2024).

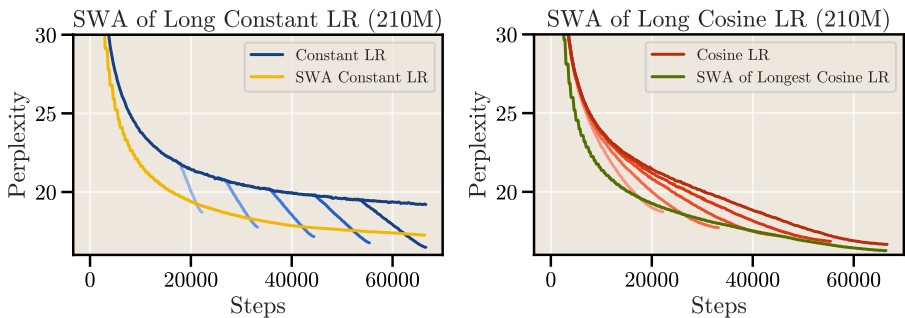

Figure 10: **SWA improves generalization and simulates decayed learning rates.** Using SWA for the constant LR phase (left) strongly boosts the loss, but a gap to the cooldown remains. SWA also improves the generalization of a cosine schedule (right), where the intermediate checkpoints of SWA largely overlap with optimal loss trajectory of shorter cosine runs.

## 4 Do We Even Need to Cooldown?

In Section 3, we show that a constant LR with a short cooldown phase can replace the cosine decay. Ideally, however, we would not need a decay at all, but aim to obtain an optimal model at any point in training, even when prolonged. This would save even more computational resources and time. In this section, we investigate two potential approaches: weight averaging and a schedule-free optimizer.

### 4.1 Stochastic Weight Averaging (SWA)

**Motivation.** While slow annealing of the learning rate can be essential to find good minima (Smith et al., 2017; Loshchilov & Hutter, 2016), Sandler et al. (2023) show theoretical and empirical equivalence between stochastic weight averaging (SWA) and cooldowns in training vision models. There, intuitively and naturally averaging reduces noise and thus improves generalization (Wortsman et al., 2022). Motivated by these results, we aim to answer the same question in LLM training: Can weight averaging replace the cooldown phase?

**Method.** We opt for a form of SWA (Izmailov et al., 2018) that splits the training in fixed windows and averages *within a window*, allowing us to keep the average as a single additional copy of the model parameters. In our experiments, we set the window to $h = 500$ steps and save the window averages as checkpoints every $h$ steps, which allows ad hoc evaluation of longer windows to see potential improvements akin to latest weight averaging (LAWA, Kaddour, 2022; Sanyal et al., 2023). For all our experiments, we find that windows below or at 2500 steps (256M tokens) are optimal. We also experimented with an exponential moving average (EMA), which performed worse than SWA, and therefore we do not report EMA.

**Experimental results.** We evaluate SWA with the same 210M model and show the results in Fig. 10 for both a constant LR (left) and cosine (right). Notably, we find a significant performance boost for SWA on top of a constant LR. Yet, it does not reach the loss values of explicit cooldowns. On the other hand, in line with previous work (Kaddour, 2022; Sanyal et al., 2023; Andriushchenko et al., 2023), we see a similar boost for SWA on top of cosine. This suggests that, regardless of schedule, SWA is a compelling approach to achieving strong models along the data-scale axis that can serve as a replacement for models trained with fewer steps, if the performance gap is acceptable and one wishes to avoid cooldowns. This is particularly advantageous as it can arguably be done *for free* on top of existing well-tuned optimizers.

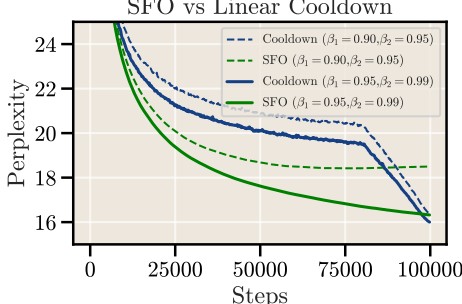

Figure 11: **The cooldown schedule outperforms SFO even when tuning momentum parameters.** We find that SFO is sensitive to the choice of $(\beta_1, \beta_2)$ momentum parameters. It gives strong performance for well-tuned momentum, but falls short of cooldown.

*Takeaway 5:* Irrespective of the schedule and without additional overhead, SWA improves performance along the training trajectory and provides better models during training. While it does not match the cooldown, it reduces the gap without the need for a separate decay phase.

## 4.2 Schedule-Free Optimizer (SFO)

Very recently, Defazio et al. (2024) introduced a schedule-free optimizer (SFO) which uses an interpolation between standard averaging and Polyak-Ruppert averaging, inspired by Nesterov's accelerated method (Nesterov, 1983). As such, the optimizer does not require a decreasing learning rate schedule, making it relevant for continual training. We seek to investigate if it can outperform the presented cooldown schedule and provide a comparison in the same LLM setting.

**Results.** We compare the results of a long 210M model run with cooldown vs. SFO with AdamW in Figure 11. Although the optimizer does not require a learning rate schedule, the authors point out that training is more sensitive to the choice of the momentum parameters $(\beta_1, \beta_2)$ (which are not identical to the momentum in Adam). They note that the optimal parameters may depend on the length of training, making it not fully schedule-free. We observe this sensitivity in our experiments, where the choice of $(0.9, 0.95)$ performs significantly worse and even increases the loss toward the end of training. For $(\beta_1 = 0.95, \beta_2 = 0.99)$, SFO performs remarkably well. Nevertheless, both settings are matched or outperformed by the cooldown schedule, in particular when comparing the same momentum configuration. We did not perform any further hyperparameter tuning for either method.

## 5 The Implications for Scaling Law Research

Our investigation shows the efficacy of a constant LR for flexible training of LLMs by introducing a short cooldown phase at any point during training. In this section, we go beyond a single model, and focus on the reliability of the alternative LR schedule compared to cosine across model sizes and scales. Ultimately, we discuss the importance for the future of scaling law experiments.

**Importance of scaling laws.** At their core, scaling laws aim to establish a functional form of a model's performance, commonly modelled as the loss $L$, as a function of the parameters $N$ or training tokens $D$; for LLMs, this is usually expressed as the power law

$$L(N, D) = \frac{A}{N^\alpha} + \frac{B}{D^\beta} + E \, ,$$

where $\{A, \alpha, B, \beta, E\}$ are variables to be estimated (Kaplan et al., 2020). Such scaling laws serve a multitude of critical purposes — from optimally spending a fixed amount of compute (FLOPs) for achieving the lowest loss, to trading off data-sources of different quality (Bi et al., 2024; Goyal et al., 2024) or modalities (Aghajanyan et al., 2023), to comparing the efficiency of different architectures (Gu & Dao, 2023; De et al., 2024).

Crucially, Hoffmann et al. (2022) demonstrated that it is necessary to vary the number of training tokens for a fixed family of models and fully decay the LR (see also Kaddour et al., 2023). At the time, their results suggested that LLMs were over-sized, leading to a substantial increase in data collection and training for much longer, beyond the Chinchilla optimal point $(N, D)$ (Touvron et al., 2023a,b).

**Why do the presented results matter for scaling?** Following the results of Chinchilla, scaling laws require a family of models each trained *from scratch* with a cosine schedule that is fit to different training lengths (cf. Section 2). In contrast, the cooldown schedule as well as weight averaging allow a much cheaper alternative in two phases: first, a model sweep with a single sufficiently long training run for each model size in the family; then, using the model checkpoints to perform a cooldown or averaging. This reduces scaling law experiments to only the model scaling axis, effectively dividing the number of necessary training runs by one order of magnitude. At the same time, it allows for flexible continual training beyond any predetermined number of steps.

**Experimental setup.** We mimic a small-scale experimental setup for scaling laws: We train a range of model sizes (33M-360M) across different token scales (0.3B-10B) on the same SlimPajama 6B dataset. For each model, we choose exactly three token counts (around the Chinchilla optimal ratio of D/N=20) in increments of 10, 20 and 30 tokens per parameter. With the cosine schedule, each model is trained from scratch three times. In contrast, for averaging, we train each model just once for the longest cycle and then use the averages at the same token count as cosine; for cooldown, we similarly take checkpoints along the constant LR trajectory and perform three annealing periods to match the token counts. We adjust the LR to be higher for smaller models. In line with the findings of Sect. 3 and for a fair comparison, we set the constant LR to be half the maximum LR for cosine and perform 20% (linear) cooldown steps. More details are given in Appendix A.1.

**Results.** We show the validation loss envelopes as a function of the training FLOPs in Fig. 12 (left) and compare each obtained model (i.e., same parameter & tokens) with cosine and its alternatives

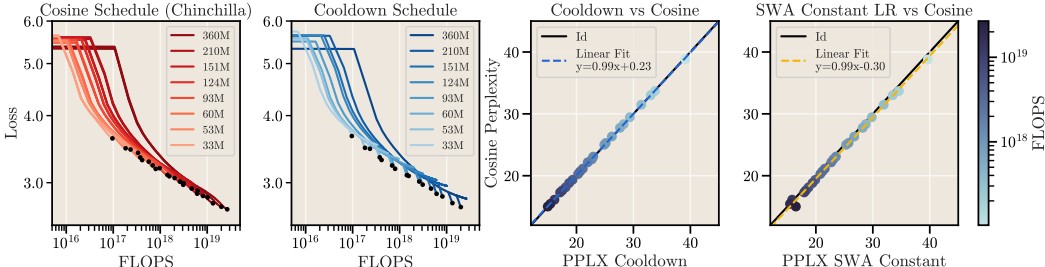

Figure 12: **Cooldown LR schedule + SWA scale reliably.** We train a range of models (33M-360M), each for three different cosine lengths. We then run each model with a constant LR just once and take the snapshots at the same token count as cosine (for SWA) or perform post-train cooldowns to the same length. Each final model is represented by a dot. **Left:** The loss curve envelopes. **Right:** The perplexity of cosine (y-axis) vs. cooldowns and SWA. Points on the diagonal indicate the same loss for both methods; above outperforming cosine, below fall short. We see alignment between both methods.

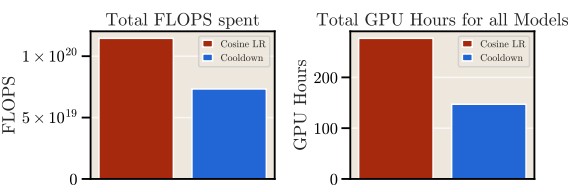

|  | **FLOPs** |
|---|---|
| **Chinchilla** | $5.59 \times 10^{23}$ |
| **w/ Cooldown** | $2.36 \times 10^{23}$ |

(a) FLOPs and GPU hours for our models.                  (b) Estimated FLOPs savings for Chinchilla.

Figure 13: **Scaling laws for a fraction of the cost.** The reliable behavior of both the cooldown schedule and SWA allows scaling experiments with a drastic reduce in both compute and GPU hours; in both our experiments (left) and the original Chinchilla (right) a factor of $\frac{1}{2}$ or less. The more training runs are performed per model size (e.g. 4 for Chinchilla), the larger the difference becomes.

(right). We find that the cooldown learning rate schedule scales *reliably*, much like cosine and SWA, to achieve optimal losses with the recipe we establish in previous sections. This is particularly visible in Fig. 12 (right), where each model's performance lies almost perfectly on the diagonal, which signals that cosine (y-axis) and cooldown (x-axis) reach the same loss after the same amount of tokens. A similar result is visible for SWA, though the reciprocal (y-offset) is negative, which agrees with our findings from Sect. 4.1 that the averaging does not fully close the gap to a LR cooldown.

**Compute savings.** Crucially, the alternatives enable scaling laws for just a fraction of the cost. In Fig. 13a, we report FLOPs and GPU hours (real wall-clock time) for all model runs for both methods. They substantially reduce both compute and GPU hours. In our experiments, where we space the runs to use token ratios of $10, 20$ and $30$, it saves half the time and FLOPs, enabling scaling laws for only a fraction of the previous cost. We report the detailed savings for all models in Appendix B.2.

**Estimating savings for Chinchilla.** We take our analysis further and estimate how much cheaper the Chinchilla model suite would have been if performed with 10% cooldowns after a single run for each model size using the model configurations as reported in Table A9 (Hoffmann et al., 2022). Since the authors do not report exact training configurations, we consider a sequence length of 1024, a batch size of 0.5M tokens and token ratios $M = D/N \in \{10, 15, 20, 25\}$ for each model. With this, we arrive at roughly $5.59 \times 10^{23}$ total FLOPs originally vs. $2.36 \times 10^{23}$ (Figure 13b). This means that less than half the compute could have been used.

> *Takeaway 6:* Scaling experiments can be done with significantly reduced compute and GPU hours by utilizing fewer but reusable training runs with a constant learning rate and ad-hoc cooldowns.

**Additional results.** We plot the training curves of all models in Appx. B.3. In addition, we show the same findings with experiments on OpenWebText2 in Appx. B.4.

# 6   Limitations

We conduct our experiments on models of up to 8B parameters, with long training runs of a 1B model on multiple hundred tokens. The trends we find are consistent across all scales, but training

behavior can be more brittle at modern scales and extremely long training (Wei et al., 2022; Tay et al., 2021). However, instabilities arising from a high learning rate for a large part of training can be alleviated (Wortsman et al., 2024).

# 7   Related work

**Cosine Schedules and Alternatives for Transformers.** The cosine decay was originally introduced by Loshchilov & Hutter (2016) for cyclic schedules in vision tasks, where it is common practice to have stepwise or cyclic LRs to escape bad minima when training multiple epochs (Smith et al., 2017). For language models where data is more vast, the cosine schedule with a single cycle is currently the de-facto standard for training, with few exceptions of T5 (Raffel et al., 2020) and PaLM1 (Chowdhery et al., 2023) that used a form of inverse square root. In line with our work, recently released models opt for alternatives such as stepwise schedules (Bi et al., 2024) or the presented constant + cooldown (Shen et al., 2024; Hu et al., 2024). These alternatives were previously also explored for vision transformers by Zhai et al. (2022), who find reciprocal square-root with cooldown to perform best, and in the context of continual learning (Ibrahim et al., 2024; Gupta et al., 2023). Defazio et al. (2023) investigate the gap between theory and practice of LR schedules and suggest that a linear decay is optimal, also for LLM training. We similarly find that a long linear decay can slightly outperform cosine.

**Weight Averaging.** Weight averaging over past iterates (Polyak & Juditsky, 1992) has long been known to be beneficial for convergence. Izmailov et al. (2018) introduce stochastic weight averaging for better generalization in deep learning models. Similarly, the exponential moving average is commonly used in vision (Morales-Brotons et al., 2024). Importantly, Sandler et al. (2023) show the equivalence of WA with decaying learning rate schedules. In the context of LLM training, close to our work is Sanyal et al. (2023) which showed that a form of latest averaging (Kaddour, 2022) can be used to improve the performance of models early in training. However, they did not investigate the relation of weight averaging to compute optimality and its implications for scaling experiments.

**Scaling Law Experiments for Neural Language Models.** Kaplan et al. (2020) were the first to establish scaling laws for language models. Important to our work, Hoffmann et al. (2022) revise these findings and demonstrate specific methods to establish scaling laws, notably training a family of models for different cosine lengths. The subsequent models like LLama1/2 (Touvron et al., 2023a,b) further improve performance of smaller models by training beyond the Chinchilla optimal point, motivated by lower inference costs (Gadre et al., 2024; De Vries, 2023; Sardana & Frankle, 2023). Recent works (Muennighoff et al., 2023; Bi et al., 2024; Goyal et al., 2024) highlight how data repetition and quality affect the scaling behavior, which suggests that scaling laws should be updated more frequently. However, these works do not consider efficient experiments for scaling laws, which is the focus of our work.

In concurrent work, Porian et al. (2024) and Pearce & Song (2024) find that the discrepancy between the Kaplan and Chinchilla scaling laws is not attributed to the learning rate decay, but they establish that the optimal token/parameter ratio can be obtained with a constant learning rate without any cooldown; however, the actual performance is then suboptimal, and a cooldown schedule as suggested in our work is needed to properly estimate model performance, in particular for downstream tasks.

# 8   Conclusion

We have demonstrated the reliability of an alternative learning rate schedule to replace cosine for LLM training, which uses a constant rate with a cooldown phase. Across a multitude of experiments, we analyze different recipes for the decay form and length. Importantly, we do not claim to have established the best learning rate schedule — instead, we investigate and demonstrate how an arguably simple recipe can match the performance of the current best practice of cosine, and discuss how it provides compelling advantages such as continual training and a strong reduction in costs for scaling law research. In addition, we find that SWA can give reliable (strong, but not optimal) estimates of models during runs, without additional overhead or training.

We believe the results are of great importance to the present and future of LLM training: The presented methods facilitate research for the current post-Chinchilla era, where models are trained much beyond compute-optimal, by allowing more flexibility to continue training whenever needed. At the same time, recent results that suggest data dependency in scaling (Bi et al., 2024; Goyal et al., 2024; Aghajanyan et al., 2023; Pandey, 2024) imply the need to frequently update scaling laws, which is economically more feasible with reduced costs. We therefore hope that our work will make scaling research more accessible to researchers and practitioners alike.

## Acknowledgements

We thank Anastasia Koloskova, Amirkeivan Mohtashami and Thomas Wolf for helpful discussions regarding the paper and its implications. We also thank Stella Biderman, whose open discussions[2] inspired parts of our experiments.

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

# A Experimental Details

## A.1 Overview

**Architecture and training parameters.** We implement the most commonly used decoder-only architecture with SwiGLU activations (Shazeer, 2020), RoPE embeddings (Su et al., 2024), RMSNorm (Zhang & Sennrich, 2019) and alternating attention and MLP blocks. Unless otherwise noted, we follow standard practices in LLM training and use the AdamW optimizer with beta parameters $(\beta_1, \beta_2) = (0.9, 0.95)$, decoupled weight decay of 0.1 (Kingma & Ba, 2014; Loshchilov & Hutter, 2017) and gradient clipping with 1.0. For warmup steps, we use a short warmup of 300 steps for the majority of runs and $1000 - 3000$ for longer runs (above 100k total steps). The cosine schedule decays the learning rate to 10% of the maximum learning rate. For most of our experiments, we use a batch size of 200, i.e., roughly 0.1M tokens for a sequence length of 512. The vocabulary is based on the GPT-2 tokenizer (Radford et al., 2019) and contains 50304 tokens.

**Dataset and evaluation.** Our main body focuses on results using SlimPajama (Soboleva et al., 2023), a cleaned and deduplicated corpus that includes webcrawl, code, papers and other sources, which is commonly used for pretraining LLMs. We use a subset of the full corpus that comprises roughly 6B tokens and randomly sample a validation set of roughly 3M tokens. During training, we evaluate the models with a fixed set of 32 batches of sequence length 512 (the same context length as training) to establish validation loss curves. At the end of training, we compute the full validation set perplexity. We perform further experiments that verify our main findings with OpenWebText2 (Gao et al., 2020) in Appx. B.4.

**Implementation and infrastructure.** Our code is based on an extension of NanoGPT[3] and uses PyTorch (Paszke et al., 2017) as well as FlashAttention (Dao et al., 2022). We incorporate bfloat16 for memory and throughput, trained with mixed precision float32 parameters and bfloat16 activations (Micikevicius et al., 2017). All experiments (aside from 1B and 8B, see A.2) were performed using a cluster of A100 GPUs (both 40GB/80GB RAM) with 2 data-parallel (i.e. 2 GPUs per run). Some selected runs used a single node of 8 H100s. We estimate that the full cost of all experiments for this project (including prototyping) to amount to roughly 2500-3000 GPU hours.

**Model configurations.** We provide an overview of the model sizes and configurations in Table 1 and the parameters for training and length in Table 2. All models for the scaling law experiments are trained with 300 warmup steps and a sequence length of 512.

| Model Size | d_model | n_layers | ffw_size | kv_size | n_heads |
|---|---|---|---|---|---|
| 33M | 384 | 8 | 1024 | 64 | 6 |
| 53M | 512 | 8 | 1536 | 64 | 8 |
| 60M | 512 | 10 | 1536 | 64 | 8 |
| 93M | 640 | 12 | 1792 | 64 | 10 |
| 124M | 768 | 12 | 2048 | 64 | 12 |
| 151M | 768 | 16 | 2048 | 64 | 12 |
| 210M | 768 | 24 | 2048 | 64 | 12 |
| 360M | 1024 | 24 | 2816 | 64 | 16 |

Table 1: **Model configurations for scaling law experiments.** We provide an overview of the model sizes and hyperparameters for the different models in the scaling experiments.

## A.2 Setup for the 1B and 8B Runs

**Setup.** In order to efficiently scale to larger models, we use the `nanotron` library [4] that enables all forms of training parallelism while being performant and flexible. The configuration and hyperparameters of the two models are described in Table 3. Similarly to our other experiments, we use the most common setup of the AdamW optimizer with $(\beta_1, \beta_2) = (0.9, 0.95)$, weight decay of 0.1,

---

[3]`https://github.com/karpathy/nanoGPT`
[4]`https://github.com/huggingface/nanotron`

and gradient clipping with 1.0. For the 1B model, we use a custom BPE tokenizer[5] (that is very similar to GPT-2) because the FineWeb dataset (Penedo et al., 2024) was already pre-tokenized on our cluster. Each run for the 1B model was performed on 4xH100s GPUs. For the 8B model, we reuse the original Llama3 tokenizer (Dubey et al., 2024) for the similar reason that FineWeb-Edu was pre-tokenized on our cluster; here, we use 12 nodes, each composed of 4xGH200 GPUs. The model is split within one node with tensor parallel (TP) set to 4.

**Evaluation.** For evaluation, we use the `lighteval` library (Fourrier et al., 2023) that works directly with nanotron checkpoints. We run the most common LLM benchmarks including MMLU (Hendrycks et al., 2021), ARC (Clark et al., 2018), OpenBookQA (Mihaylov et al., 2018), PIQA (Bisk et al., 2020), HellaSwag (Zellers et al., 2019), CommonSenseQA (Talmor et al., 2019), SIQA (Sap et al., 2019), Winogrande (Sakaguchi et al., 2021), truncating large benchmarks to 1000 samples so that we could efficiently evaluate over the course of training. This is a custom setup that is equal to that used for FineWeb ablations; the setup to reproduce the eval is described here[6] [7]. We report the accuracy normalized by sequence length (`acc_norm`).

| Model | LR (Cos/Const) | BS | Steps | Tokens | Token/Params Ratio |
|-------|----------------|-----|-------|--------|--------------------|
| 33M | (2e-3, 1e-3) | 0.1M | [3k, 7k, 10k] | [0.3B, 0.7B, 1.0B] | [9.2, 21.4, 30.6] |
| 53M | (2e-3, 1e-3) | 0.1M | [3.7K, 7.5K, 11.2K] | [0.4B, 0.8B, 1.2B] | [7.2, 14.5, 21.7] |
| 60M | (2e-3, 1e-3) | 0.1M | [7.5K, 12.5K, 17.5K] | [0.8B, 1.3B, 1.8B] | [12.8, 21.4, 30.0] |
| 93M | (2e-3, 1e-3) | 0.1M | [10K, 17.5K, 25K] | [1.0B, 1.8B, 2.6B] | [11.0, 19.2, 27.5] |
| 124M | (1e-3, 5e-4) | 0.1M | [15K, 25K, 35K] | [1.5B, 2.6B, 3.6B] | [12.4, 20.7, 29.0] |
| 151M | (1e-3, 5e-4) | 0.1M | [25K, 37.5K, 50K] | [2.6B, 3.8B, 5.1B] | [16.9, 25.3, 33.7] |
| 210M | (1e-3, 5e-4) | 0.1M | [37.5K, 50K, 62.5K] | [3.8B, 5.1B, 6.4B] | [18.4, 24.6, 30.7] |
| 360M | (1e-3, 5e-4) | 0.2M | [25K, 37.5K, 50K] | [5.1B, 7.7B, 10.2B] | [14.2, 21.3, 28.5] |

Table 2: **Training parameters for scaling experiments.** We describe the learning rates, training lengths and ratios for the different models in the scaling experiments.

| Parameter | 1B | 8B |
|-----------|-----|-----|
| **d_model** | 1792 | 4096 |
| **n_layers** | 24 | 32 |
| **ffw_size** | 4864 | 14336 |
| **kv_size** | 128 | 128 |
| **n_heads** | 14 | 32 |
| **n_kv_heads** | 14 | 8 |
| **vocab_size** | 49152 | 128256 |
| **seq_len** | 2048 | 4096 |
| **warmup_steps** | 2000 | 1000 |
| **batch_size** | (1.8M, 2M) | 0.6M |
| **Total Steps** | (55k, 220k) | 20k |
| **Peak LR** | 8e-4 | 3e-4 |

Table 3: **Model configurations for larger runs.** The batch size and learning rate for the 1B model tokens were estimated using DeepSeek scaling laws for 100B tokens. The two values for BS and the total steps of the 1B model distinguish the runs (100B,460B). For the 8B model, the architecture is identical to Llama3 and the batch size was set according to the available GPU limits on our cluster at the time of running experiments.

---

[5]`https://hf.co/lvwerra/the-tokenizer-v1`
[6]`https://github.com/huggingface/cosmopedia/blob/d62abb8e13c567b999e33d0a1d795968bc052c6a/evaluation/README.md`
[7]`https://hf.co/datasets/HuggingFaceFW/fineweb/blob/main/lighteval_tasks.py#L12`

## A.3 FLOPs computation

We do not rely on the heuristic of 6=ND for computing the FLOPs of a Transformer model, but use a more thorough computation that involves the embeddings, attention and MLP operations directly. For reproducibility and other researchers to use, we provide our Python code in Figure 14.

```python
def embedding(seq_len, vocab_size, d_model):
    return 2 * seq_len * vocab_size * d_model

def attention(seq_len, d_model, key_size, num_heads):
    projections = 2 * 3 * seq_len * d_model * (key_size * num_heads)
    logits = 2 * seq_len * seq_len * (key_size * num_heads)
    softmax = 3 * num_heads * seq_len * seq_len
    softmax_query_reduction = 2 * seq_len * seq_len * (key_size *
        num_heads)
    final_layer = 2 * seq_len * (key_size * num_heads) * d_model
    return projections + logits + softmax + softmax_query_reduction +
        final_layer

def dense(seq_len, d_model, ffw_size, swiglu=False):
    if swiglu:
        return 2 * seq_len * (3 * d_model * ffw_size)
    else:
        return 2 * seq_len * (2 * d_model * ffw_size)

def final_logits(seq_len, d_model, vocab_size):
    return 2 * seq_len * d_model * vocab_size

def flops(
    n_layers,
    seq_len,
    vocab_size,
    d_model,
    key_size,
    num_heads,
    ffw_size,
    swiglu=True,
):
    flops_single = (
        embedding(seq_len, vocab_size, d_model)
        + n_layers
        * (
            attention(seq_len, d_model, key_size, num_heads)
            + dense(seq_len, d_model, ffw_size, swiglu=swiglu)
        )
        + final_logits(seq_len, d_model, vocab_size)
    )
    # assume backward pass has twice the FLOPs of the forward pass
    return 3 * flops_single
```

Figure 14: **FLOPs computation.** Instead of the common approximation of 6=ND, we use more detailed calculations for the FLOPs estimation based on the Transformer model configuration. We provide the Python code above.

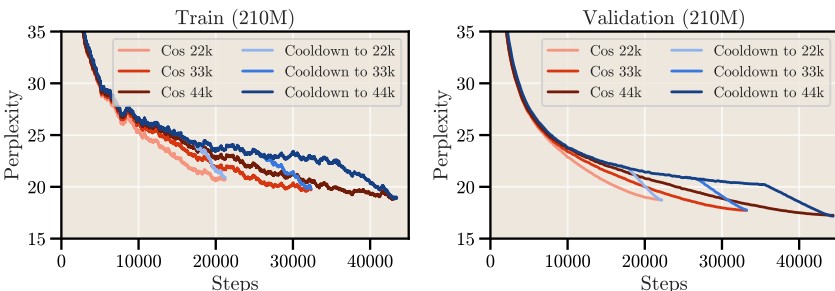

Figure 15: **The difference in loss curves of cosine vs. constant learning rate with cooldown.** The cooldown phase initiates a sharp decrease in loss for both training (left) and validation (right). The training perplexity is averaged out over a window to achieve a smoother curve. For validation, we use a fixed set of 32 validation batches to report the loss each step, which creates a smooth curve by design.

# B  Additional Results

## B.1  More Results on Cooldown

**Ablation of functional form of cooldown.** We provide additional experiments to show that our results are consistent and robust to different changes. For simplicity, we only cooldown to 50K steps here, but we found these results hold regardless of the step at which we decay. In Fig. 16 and Fig. 17, we test different functions for the cooldown phase:

- Linear: Classical linear decay.
- 1 - Sqrt: Defined in Eq. (1-sqrt).
- Cosine: Similar to the classical cosine decay, but applied only during the decay phase.
- Mirror Cosine: The symmetric counterpart of the cosine function with respect to the linear decay.
- 1 - Square: The square root function in Eq. (1-sqrt) is replaced by a square function.

Note that the order might change for substantially different cooldown lengths; we focus on 10% and 20% because they are practically relevant.

In addition, we perform a comparison of different exponents for the square-root function: since the (1-Sqrt) can be expressed as $(1 - x^a)$ where $a = 0.5$, we sweep the parameter $a < 0.5$. The results are shown in Figure 18. Apart from 0.1 and 0.2, which perform noticeably worse because the learning rate is too low for many steps, the other exponents only show a marginal difference to $a = 0.5$, which still comes out on top.

We aim to further investigate the impact of the functional form of the cooldown in future work.

**Length of cooldown.** We repeat our ablation from Sect. 3 with a 210M model in Figure 19 (fractional x-axis) and with the absolute number of decay steps (not fractional) in Figure 20.

**LR sensitivity.** The optimal learning rate also transfers to different cooldown lengths in our experiments, see the results in Figure 21.

**Annealing cosine to less than 10%.** We ablate the choice of the final learning rate for the cosine annealing in Fig. 22, where we find that the final value should be set lower than simply 10% of the maximum. However, when evaluating on downstream benchmarks (see Section B.5), we see that annealing to zero can hurt metrics; we posit this comes from too early saturation. Combining these two arguments implies that the maximum and final LR should be set (and ideally swept over) independently.

## B.2  Additional Results and Compute Savings

We give the savings for all models used in the scaling experiments in terms of FLOPs in Figure 24 and GPU hours in Figure 25.

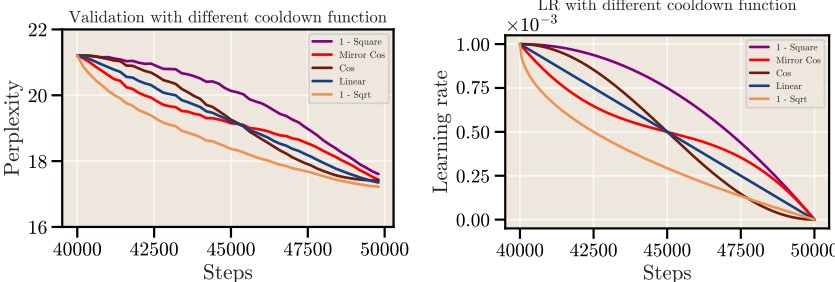

Figure 16: **Different cooldown functions.** We test various cooldown schedule functions and consistently observe the same order of performance (left), with (1-sqrt) being the most effective. Remarkably, the drop in loss closely follows the learning rate (right) even for different functions.

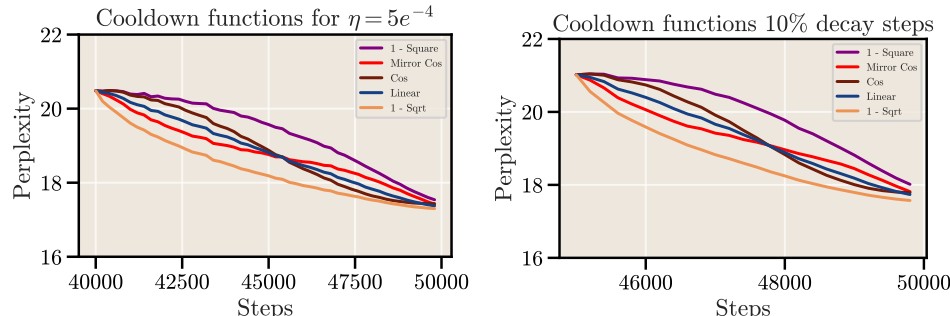

Figure 17: **The order and results of different cooldowns hold across settings.** Left: We change the learning rate to $5e^{-4}$ instead of $1e^{-3}$ as in previous experiments. Right: The performance for 10% cooldown steps instead of 20%. Note that the order might change for substantially different cooldown lengths; we focus on 10% and 20% as they are practically relevant.

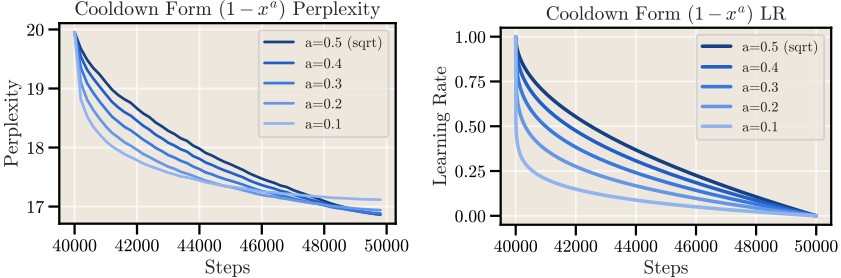

Figure 18: **Different exponents for functions.** Since the negative square root performs remarkably well, we experiment with varying the exponent of the functional form $(1 - x^a)$ for $a < 0.5$. Besides 0.1 and 0.2 which perform noticeably worse, the other exponents only show marginal difference in this experiment, with $a = 0.5$ (the square root) still coming out on top.

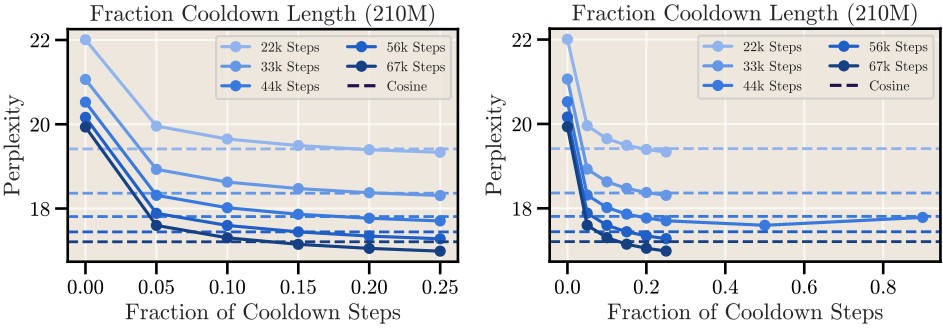

Figure 19: **Parabola shape of the relationship between cooldown length and final perplexity.** We repeat the experiment from Fig. 5 with a 210M parameter model and the zoomed-in view on the left.

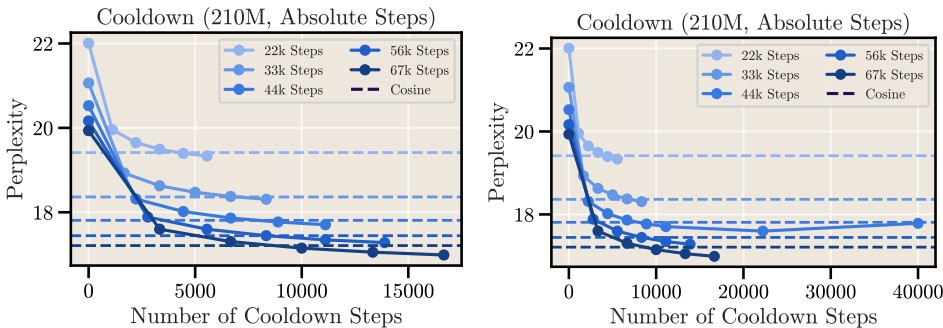

Figure 20: **The effect of the cooldown length in terms of absolute steps.** We repeat the plots from Fig. 19 with the absolute number of steps on the x-axis.

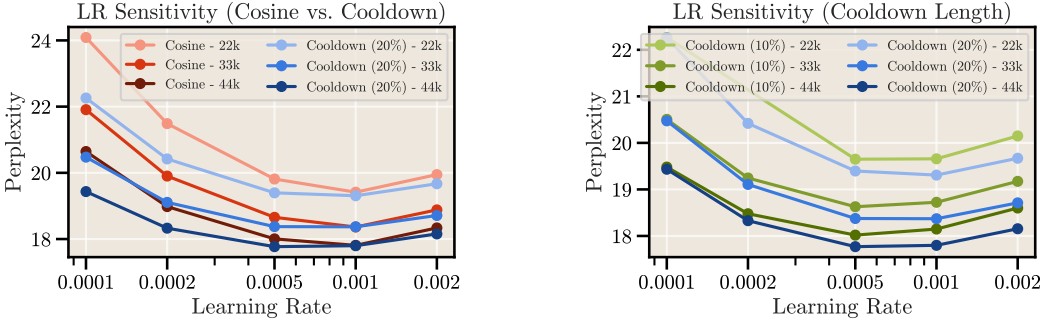

Figure 21: **Learning rate sensitivity for cosine and cooldown with different lengths.** The optimal learning rate also transfers to different cooldown lengths.

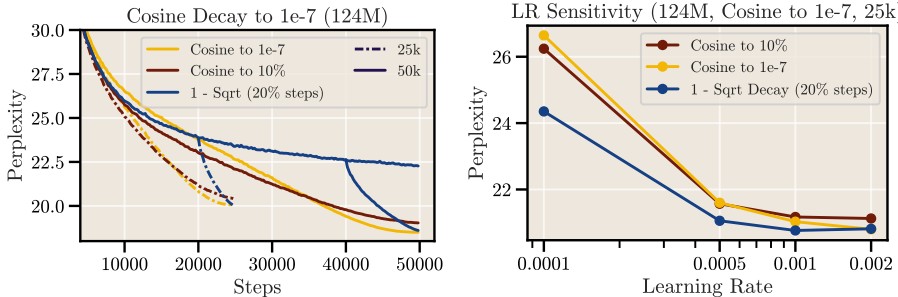

Figure 22: **The cosine schedule should be decayed to lower values.** While setting the minimum to 10% of the maximum LR is the most common heuristic, the final LR should be set lower and independently of the maximum. In our experiments, cosine then matches the cooldown with (1-Sqrt) (left; optimal LR shown), albeit the LR sensitivity remains (right; larger maximum LR led to divergence).

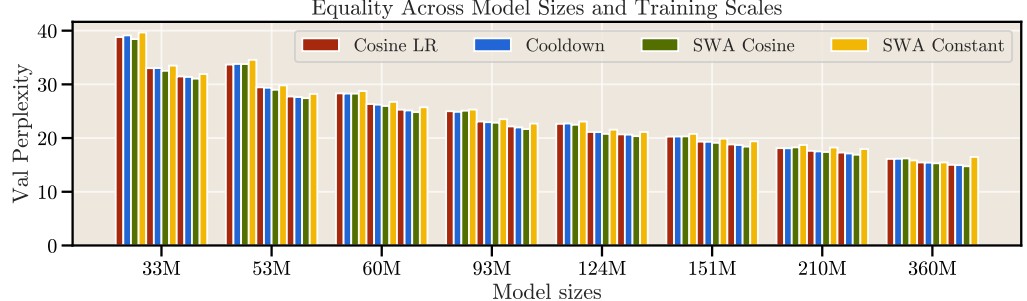

Figure 23: **Final validation perplexity of all models in scaling experiments across different runs.**

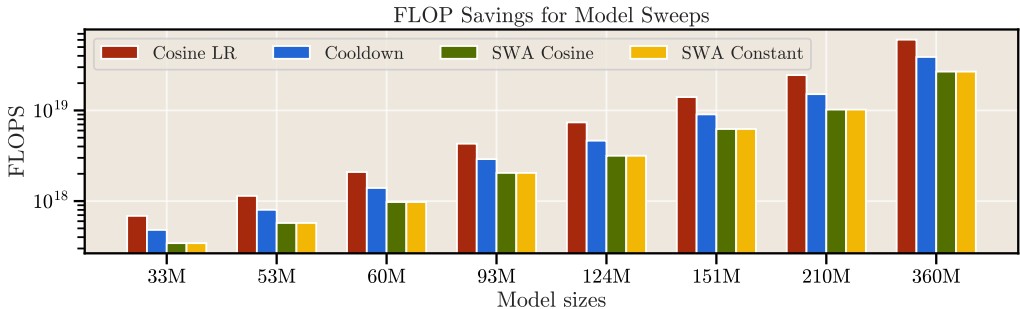

Figure 24: **FLOPs savings for all models in scaling experiments across different runs (log scale).** The savings amount to a factor of $\frac{1}{2}$ across all models. If more runs across different lengths were to be performed (compared to 3 per model in our experiments), the difference would be even more significant.

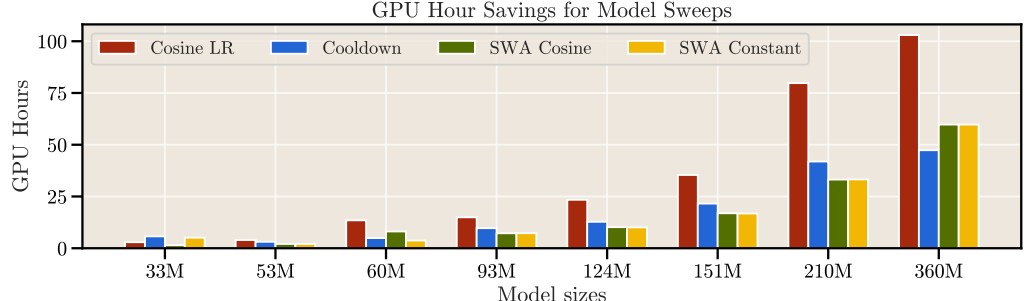

Figure 25: **GPU hours savings for all models in scaling experiments across different runs.** We report the actual runtimes summed up over the sweeps for all models. The savings in terms of runtime become especially more prominent for bigger models and longer training runs. Please note that the hours for SWA of the 360M model are slightly off because of congestion in our cluster during the runs.

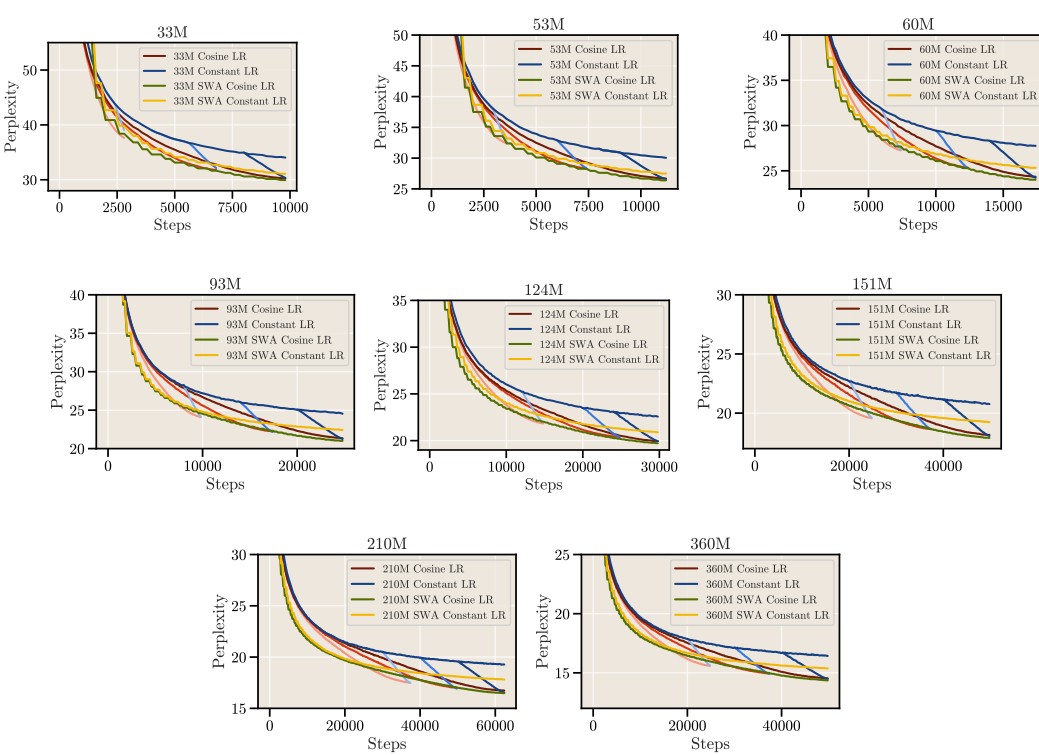

Figure 26: **Validation Loss Curves (Perplexity) of all Models in Scaling Experiments.** We visualize all training runs for the models used in the scaling experiments in Section 5.

### B.3 Learning Curves for All Models of Scaling Experiments

We provide the learning curves for all models in the scaling experiments in Figure 26. The final validation perplexity for all models and methods is given in Figure 23.

### B.4 Additional Experiments on OpenWebText2

In addition to all results on SlimPajama, we perform experiments on the commonly used benchmark of OpenWebText2 (Gao et al., 2020) with models of sizes 60M, 93M and 166M. As shown in

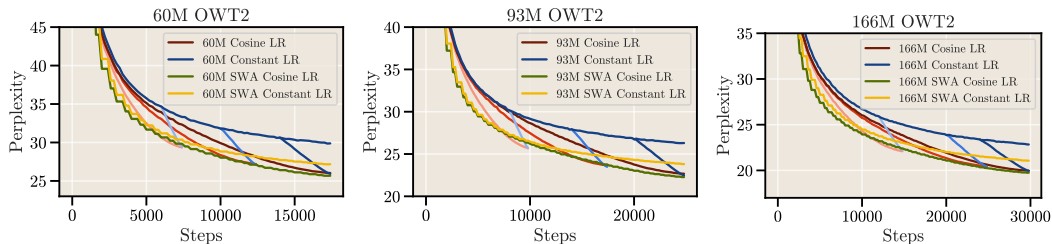

Figure 27: **Results Transfer to OpenWebText2.** We see the same behavior for cooldown schedules and SWA, verifying the reliability of our findings.

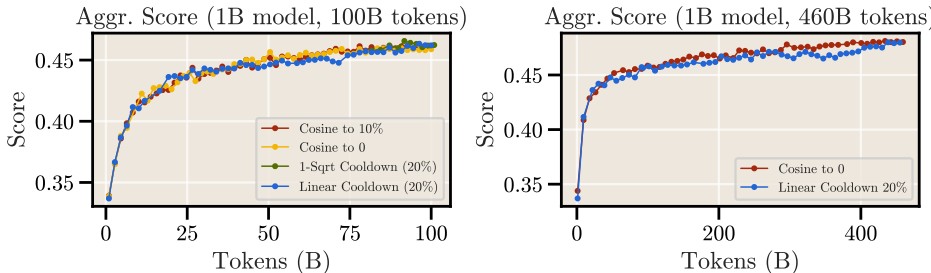

Figure 28: **Aggregate metrics throughout training of the 1B model on 100B and 460B tokens.** We train a 1B model on 100B (left) and 460B tokens (right) of FineWeb (Penedo et al., 2024), and find that the performance of cosine and cooldown matches. Though cosine to zero improves the loss (Figure 22), it leads to a saturation before the end of training, hurting overall performance.

Figure 27, our findings from previous experiments succesfully transfer, where the cooldown recipe matches the performance of cosine and SWA boosts performance during training.

## B.5 Full Results of Large Model Runs

In this section, we report the detailed results of the 1B runs with the downstream benchmarks.

In Figure 28, we compare the aggregate metrics throughout training for both the 100B (left) and the 460B token run (right). We equally plot individual benchmark curves in Figure 29 for the 100B run with a zoomed view after 80B tokens in Figure 30; interestingly, we observe a similar uptick in performance for some metrics (e.g., MMLU, HellaSwag) with the cooldown, while others do not benefit as clearly (e.g., OpenBookQA). This observation is an interesting direction for further research.

The final numbers are given in Table 4 (100B) and Table 5 (460B). Notably, longer cooldowns do not necessarily improve the metrics (e.g. going from 5% to 20%).

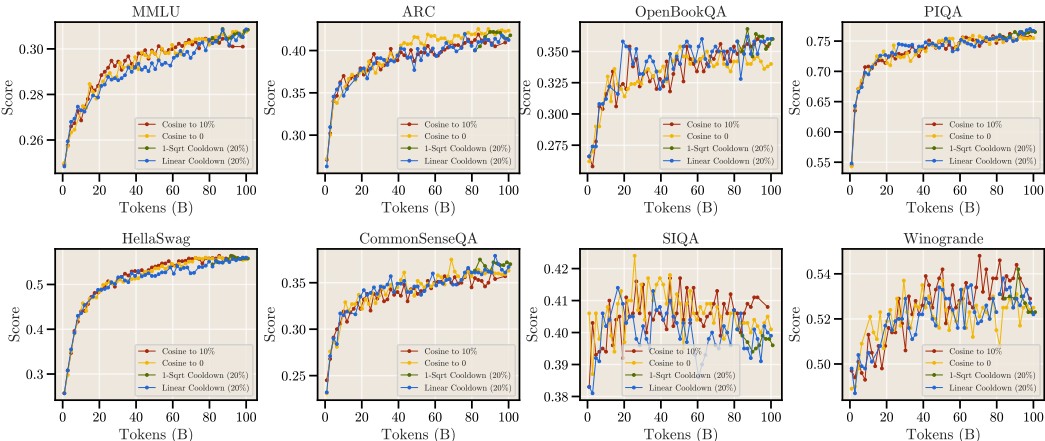

Figure 29: **Detailed benchmarks throughout training of the 1B model on 100B tokens.** For the cooldown (starting at 80B tokens), we observe a similar uptick in performance for some metrics (e.g., MMLU, HellaSwag) akin to the observed drop in loss. Other metrics do not benefit as clearly from the cooldown (e.g., OpenBookQA).

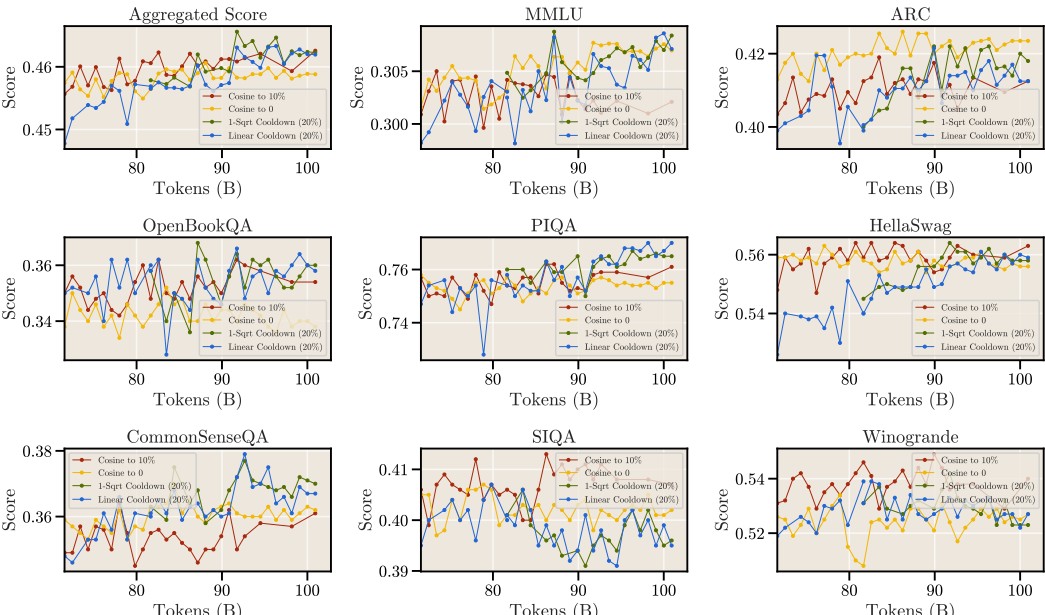

Figure 30: **Zoomed-in view of the 1B model benchmarks after 80B tokens.** We repeat Figure 29 with a focus on the cooldown phase, which starts at 80B tokens.

| Metric | Cosine to 10% | Cosine to 0 | 1-Sqrt 20% | Linear 20% |
|---|---|---|---|---|
| **Aggregated Score** | 46.26 | 45.88 | 46.23 | 46.20 |
| **MMLU** | 30.21 | 30.70 | 30.84 | 30.71 |
| **ARC** | 41.25 | 42.35 | 41.80 | 41.25 |
| **OpenBookQA** | 35.40 | 33.80 | 36.00 | 35.80 |
| **PIQA** | 76.10 | 75.50 | 76.50 | 77.00 |
| **HellaSwag** | 56.30 | 55.60 | 55.80 | 55.90 |
| **CommonSenseQA** | 36.10 | 36.20 | 37.00 | 36.70 |
| **SIQA** | 40.70 | 40.20 | 39.60 | 39.50 |
| **Winogrande** | 54.00 | 52.70 | 52.30 | 52.70 |

Table 4: **Final evaluation results after 100B tokens.** Both cosine and the cooldown schedules have comparable final numbers, with only slight differences for certain benchmarks.

| Metric | Cosine to 0 | 1-Sqrt 5% | Linear 5% | Linear 10% | Linear 20% |
|---|---|---|---|---|---|
| **Aggregated Score** | 48.03 | 47.91 | 47.84 | 47.98 | 47.92 |
| **MMLU** | 31.25 | 31.71 | 31.65 | 31.78 | 31.84 |
| **ARC** | 45.20 | 44.10 | 44.05 | 44.15 | 45.05 |
| **OpenBookQA** | 37.60 | 37.80 | 38.00 | 38.20 | 37.20 |
| **PIQA** | 78.10 | 77.40 | 77.40 | 77.30 | 77.80 |
| **HellaSwag** | 59.90 | 59.60 | 59.50 | 60.00 | 59.20 |
| **CommonSenseQA** | 37.70 | 37.30 | 36.70 | 36.90 | 36.90 |
| **SIQA** | 39.90 | 39.50 | 39.40 | 39.50 | 39.70 |
| **Winogrande** | 54.60 | 55.90 | 56.00 | 56.00 | 55.70 |

Table 5: **Final evaluation results after 460B tokens.** The findings of Table 4 transfer to much longer training runs with 460B tokens, where the performances of cosine and cooldowns match well. Notably, longer cooldowns do not necessarily improve the metrics (e.g. going from 5% to 20%).

