# OpenReview forum: "Scaling Laws and Compute-Optimal Training Beyond Fixed Training Durations"
_NeurIPS.cc/2024/Conference — NeurIPS 2024 spotlight_

### Official Review · Reviewer_yzXN · 2024-06-24

**Soundness:** 3
**Presentation:** 2
**Contribution:** 3
**Rating:** 5
**Confidence:** 4

**Summary:**

The authors present an empirical study on using constant learning rates (plus a short cooldown) instead of cosine schedules for training LLMs. The authors show that:

- Constant LR + cooldown roughly matches cosine schedules. (Fig 3,4)
- SWA of a long schedule almost matches the performance of shorter schedule given the same number of steps. (Fig 8).
- Chinchilla-type scaling laws can be derived with constant LR schedules + cooldowns which will save compute. (Fig 10)

**Strengths:**

- The potential impact of constant LR instead of cosine is high.
- The paper is rather well written.

**Weaknesses:**

- The experimental setup is unclear. But this should be very easy to fix in an updated version.
- It is still unclear to me exactly how well the match between constant LR and cosine is, especially for longer training.
- The results on SWA and constant LR seems almost orthogonal.
- Higher performance is not reached, so the only upside is in doing scaling studies. This might only be relevant for a few well-funded labs.

**Questions:**

1. Why plot perplexity instead of val loss?
2. Could you have a subsection called experimental setup? On line 104 you mention “. We follow our experimental setup and train a 210M model”, but I don’t think the details of this model is given earlier.
3. Based upon Figure 2, it seems like your schedule goes to 0 whereas cosine typically does not go to 0. Is this a confounder in your experiments?
4. Fig 4 shows constant LR beating cosine while Fig 3 shows them matching. Why this discrepancy?
5. Fig 6 (right) seems to show that there is still a small gap between constant (1-sqrt) and cosine – is this correct?

---

> ### Author Rebuttal · Authors · 2024-08-06
>
> Thank you very much for taking the time to read our work and the detailed questions. You can find the replies below:
>
> > **On the experimental setup and perplexity (weaknesses and Questions 1-2):**
>
> Thank you for the feedback -- we have made the setup and pointers to the exact details more clear in the updated version of the paper. In essence, we follow the most common setup of LLM pretraining with a decoder-only Llama architecture and a large web scale dataset. In the submitted version, we describe this setup in the background Section 2 and follow it for the rest of the paper, with exact details given in Appendix A.1.
> Regarding the metrics, we opt for the perplexity instead of the loss simply because it makes interpreting results easier, as the values are spread further apart.
>
> > **The match between cooldown and cosine (from weaknesses):**
>
> All our experimental results point towards a strong match between cosine and the constant LR when the cooldown is performed between 10-20% of training duration. In particular, this is the goal of Section 5 and Figure 10, where each schedule leads to the same result for the same model and training length.
> Notably, we have been able to verify this alignment with our larger scale runs of a 1B model using downstream evaluations; see our general response above.
>
>
> > **Results on SWA and constant LR (from weaknesses):**
>
> There is a strong theoretical and empirical connection between weight averaging and learning rate decay (see [1]). We were particularly motivated to potentially replace the cooldown with SWA entirely; while it does not fully match the performance, we found a strong improvement that has practical relevance and merits a mention in the paper. Moreover, we believe these insights might inspire future work in this direction, e.g., cleverly combining SWA and cooldowns.
>
> > **Relevance only for large labs (from weaknesses):**
>
> We agree that scaling studies are compute intensive and therefore most accessible to well-funded labs. However, the aim of our work is precisely to reduce the GPU hours required to train models or run scaling studies. This is possible by making it easier to 1) continue training a model for more tokens without restart, and 2) see a model at different scales of compute via checkpoints and ad-hoc cooldowns. We therefore hope that our findings enable (and encourage) smaller labs to perform scaling studies.
>
>
> > **The scale of the cooldown (Question 3):**
>
> You are right that cosine does not typically go to zero, whereas the LR cooldown does. We have therefore added new ablations to investigate this.
> While going to zero helps cosine achieve lower loss values, we find that it overall hurts benchmark metrics. We posit that annealing to zero with cosine leads to a saturation before the end of training, as large portions will be spent with very low LR values.
> Please find the detailed results in our general response to all reviewers above.
>
> > **Difference between Figure 3, 4 and 6 (Questions 4-5):**
>
> We apologize for the confusion. Figure 3 shows the match between cosine and a linear cooldown for 20%, where we sweep the peak LR and plot the optimal one for both approaches.
> Figure 4, on the other hand, shows the curves with the same LR (not necessarily optimal) that was found in the previous experiment for much longer runs (200k steps instead of 45k), where we find that the cooldown (again for 20% of steps) can outperform cosine.
> In contrast, Figure 6 shows a cooldown for only 5% of training steps. This was motivated by our ablation on the cooldown length in Figure 5. Importantly, we find a negligible gap between cosine and such a short cooldown (compared to the 20% used before).
> Following your feedback, we have adapted the captions to make this distinction clearer.
>
> ---
>
> Please let us know if this response clarifies your questions or if you would like to discuss more points.
>
>
> [1] Sandler, M., Zhmoginov, A., Vladymyrov, M., and Miller, N. Training trajectories, mini-batch losses and the curious role of the learning rate. Jan 2023. URL http://arxiv.org/abs/2301.416 02312v2.

---

### Official Review · Reviewer_5fmf · 2024-07-10

**Soundness:** 4
**Presentation:** 4
**Contribution:** 3
**Rating:** 9
**Confidence:** 5

**Summary:**

Scaling Laws and Compute-Optimal Training without Fixed Training Duration

Summary
A major weakness of the cosine annealing learning rate schedule, one of the most prevalent learning rate schedules in LLM training, is that for optimal performance the cycle length must be adjusted based on the training duration. Intermediate checkpoints generally do not achieve optimal loss. This is inconvenient since achieving the optimal loss for various training durations requires individual runs for each duration. The authors investigative alternative scheduling methods for alleviating this difficulty. Namely, the authors seek solutions which allow provide near optimal intermediate checkpoints along one long training run, potentially with a relatively small overhead from applying cooldown or averaging. Such solutions are especially useful for computing scaling laws and hyperparameter tuning by reducing the computation cost significantly. Through careful experimentation the authors provide several insights about the following solutions (which have been proposed previously in the literature).

1.	Trapezoidal LR schedule with warmup + constant lr + cooldown
2.	Stochastic Weight Averaging
3.	Schedule Free Optimizer from Defazio et al. [1]

In particular it is demonstrated that the trapezoidal schedule is a simple and effective solution for provide near optimal checkpoints along a potentially infinitely long trajectory.

[1] Defazio, A., Mehta, H., Mishchenko, K., Khaled, A., Cutkosky, A., & others. (2024). The road less scheduled. arXiv preprint arXiv:2405.15682.

**Strengths:**

The paper discusses an important and practical topic of LLM training. Hyperparameter tuning has been established as an important aspect of LLM training but result in prohibitive costs especially when scaling. The paper discusses solutions for trying to "eliminate" the hyperparameter of training duration which is a worthy goal with a large benefit. The paper is also written and presented clearly with many experiments and interesting results.

**Weaknesses:**

The technical novelty is a bit limited because all the approaches evaluated have been proposed before in the literature (modulo some small tweaks). However, I do not view this as a major weakness since a thorough, well-presented review of these approaches is still a strong contribution.

**Questions:**

1. In figure 1 why are there no spikes? For example the n=8k curve is rewarmed every 8k iterations. Are the curves smoothed?
2. Can the SWA be clarified a bit? Is every evaluation done on the iterate average over all checkpoints in the last 500 steps? How many checkpoints does that include?
3. What about a cooldown function like (1-x^a) where a < 0.5? All the other cooldown functions tried seem to lie "above" 1 - sqrt(x).

**Limitations:**

Yes.

---

> ### Author Rebuttal · Authors · 2024-08-06
>
> Thank you very much for the positive and detailed feedback to our work!
>
> We aim to clarify your individual questions below:
>
> > **Spikes in Figure 1**
>
> For all the figures in our paper, we plot the validation perplexity of a fixed set of sequences in the dataset; this therefore entails very smooth curves as there is no batch sampling. You are right that the curve  for n=8k (dashed black) is rewarmed cyclically, which results in a strong jump in loss in the left part of the Figure. Does this answer your question?
>
> > **Clarification of SWA**
>
> Apologies that the description of SWA was not clear enough -- we have updated the paper to reflect this feedback. We perform SWA within a window of 500 steps. For example, at step 1125, we uniformly average the 125 previous steps down to 1000; at step 1500, the average contains all steps down to 1000. This is done by keeping a single running average.  We then store these averages for each window as checkpoints (i.e., at steps 1000, 1500, 2000, …).
> Storing these checkpoints allows investigating the optimal horizon by retrospectively averaging the averages and finding the minimal validation loss. This is what we use to plot the evaluation curves in Figure 8. We found that starting in the middle stages of training, the optimal horizon stabilizes around 2500 steps (i.e. 5 checkpoints).
>
> > **A cooldown of (1-x^a)**
>
> This is a very good point and an interesting comparison! We will include such an ablation in the final version of our paper. With that, we might be able to better answer the question of what the ‘optimal’ cooldown looks like.
>
> ---
>
> Please let us know if you have additional questions, we are happy to discuss more.

---

> ### Comment · Reviewer_5fmf · 2024-08-12
>
> Thanks for the replies!
>
> 1. Maybe I'm missing something but I don't see any loss jumps in the left panel of Figure 1? All the curves are smoothly decreasing.
> 2. Got it, thanks for clarifying.
> 3. Great, looking forward to seeing what comes out.
>
> After reading the discussion with other reviewers and author responses I believe the final paper will be very nice so I will increase my confidence and maintain the high score.

---

> > ### Author Response · Authors · 2024-08-13
> >
> > Thank you for your reply and nice words!
> >
> > > Maybe I'm missing something but I don't see any loss jumps in the left panel of Figure 1? All the curves are smoothly decreasing.
> >
> > We think the submitted version of Figure 1 should contain the black dashed line for a cycled schedule ('Cycle n=8k'), which shows a loss jump at steps 8k and 16k -- is the Figure different for you? The line is partially hidden under the very light red one for n=8k, which might lead to the confusion.

---

### Official Review · Reviewer_J233 · 2024-07-13

**Soundness:** 4
**Presentation:** 4
**Contribution:** 3
**Rating:** 8
**Confidence:** 5

**Summary:**

The paper focuses on learning rate schedules in scaling large language models (LLMs). Traditionally, LLMs are trained with a cosine learning rate schedule, which requires a pre-specified training duration for learning rate decay. This makes it difficult to dynamically adjust the training duration, as early stopping and continued training after the decay result in suboptimal model performance. To address this, the authors study an alternative schedule of constant learning rate with cooldown, which enables one to get the optimal model at any step during the training process. The authors systematically compare different choices of cool down duration and cool down schedule, and demonstrate that constant learning rate with 1 - sqrt cooldown schedule matches the performance of cosine schedule tuned for that duration.

Beyond cooldown, the authors compare two different approaches of removing the need of learning rate schedule: stochastic weight averaging and schedule-free optimizer. The authors show that while they exhibit promising performance, the performance of models obtained in the middle of the training still cannot match the performance of the proposed constant learning rate with cooldown.

Finally, the authors validate the proposed learning rate schedule in the scaling law study of compute optimal models, and demonstrate that the constant learning rate with cooldown schedule leads to the same results as cosine learning rate schedule, while saving half of the compute flops.

**Strengths:**

The paper presents a simple approach to address a major limitation of cosine learning rate schedule, and demonstrates its effectiveness with solid experimentations. Overall I think the paper is of very high quality.


The proposed method is simple and highly practical, which makes it relevant for a broad range of today’s scaling studies. It is widely known that scaling is the key to the success of LLMs, and it is important to establish compute optimal scaling laws as large scaling training is really expensive. The proposed learning rate schedule is able to halve the flops of the chinchilla compute optimal horizon study without adding extra implementation or tuning complexity, which I believe is a major contribution.

The proposed method is supported with solid experimentations. The authors include comprehensive comparisons of different schedules of cooldown and hyperparameter choices, which convincingly establish the advantage of the proposed schedule and can also serve as tuning guide when adopting it. The authors also compare the proposed method to popular alternative approaches.

The paper is well written. The main idea and the experiments are laid out in a logical way that makes it easy to follow. The experiment results are also presented in an intuitive way, making it easy to compare methods.

**Weaknesses:**

While the experiment results are very solid, I do hope that the authors could expand the scale of experiments to larger models. Larger models are usually more sensitive to suboptimal hyperparameters, and therefore it would be great if the proposed method can also be verified in larger models, such as a 3B or 7B model which are common sizes for practical LLMs.

**Questions:**

Another highly relevant use of training LLMs without fixed duration is continued pre-training, which is quite common in practice as people often continue the pre-training on specialized data for different use cases (such as turning general models into code specialized models). It would be great if the authors could include some discussions and experiments on how the proposed constant learning rate with cooldown can be applied in continued pre-training. For example, one question to ask would be would it be better to continue from the checkpoint before or after cooldown.

**Limitations:**

The authors have sufficiently addressed the limitations and societal impacts.

---

> ### Author Rebuttal · Authors · 2024-08-06
>
> We thank you very much for your valuable comments and kind words!
>
> > **Expansion to larger models**
>
> Please see the general response above for the results on both a 8B and a 1B model, which we were able to train without issues out of the box. This makes us confident that the match of performance of the schedules also holds for larger scales, and we aim to investigate even larger models in the future. We also look forward to more adoption and ablations in different settings of the community.
>
> > **Discussion on continued pretraining**
>
> We fully agree on the relevance to continued pretraining and have expanded our discussion on these points in the updated version of the paper.
> The natural approach is to use checkpoints before the cooldown to continue training with a high LR; this avoids loss spikes and brittle training when rewarming the learning rate (cf. Figure 1 in our paper with cosine). Moreover, rewarming has been reported to hurt performance and introduce forgetting compared to single annealing training [1], though careful strategies can alleviate such issues. We think it remains an interesting question if a single cooldown schedule is absolutely optimal given a total compute budget for LLM training.
>
> ---
>
> Please let us know if you would like us to elaborate any additional points, we’d be happy to discuss.
>
> [1] Ibrahim, A., Thérien, B., Gupta, K., Richter, M. L., Anthony, Q., Lesort, T., Belilovsky, E., and Rish, I. Simple and scalable strategies to continually pre-train large language models. Mar 2024. URL https://arxiv.org/abs/2403.08763v3

---

> > ### Comment · Reviewer_J233 · 2024-08-10
> > **Re: Rebuttal**
> >
> > I'd like to thank the authors for answering my questions and providing us with additional experiments. My concerns and questions have been fully addressed.

---

### Author Rebuttal · Authors · 2024-08-06

We thank everyone for taking the time to read our work and the positive & constructive feedback. We are very encouraged by the positive comments of the reviewers to our findings and the writing.

We also thank all reviewers for the useful comments with detailed questions, to which we reply in individual comments. We look forward to engaging in further discussion, either for questions or potential improvements.

Since the submission date and with the points raised, we have been working diligently on expanding and generalizing our results. In particular, we want to highlight **larger scale runs and investigation of evaluation metrics** to all reviewers. In summary, we have results for 1) a 8B model run, 2) evaluation benchmarks (beyond loss) on a 1B model for 100B and 460B tokens, and 3) ablations of cosine to zero at both small and large scale.

## 1) Validation on a 8B run
We have been able to validate the LR schedule for a **8B model (Llama 3.1 architecture)**, showcasing the exact same behavior as we have demonstrated on smaller scales. We rely on the same peak LR as Llama3 (0.0003) with a smaller batch size (~0.3M tokens) and train on FineWeb-Edu [3]. In Figure 1 in the attached PDF, we show the loss curves for both cosine and the constant LR (20% cooldown with 1-Sqrt) and find that the **training is stable** and **performance matches**. Albeit it is a short run (6B tokens total), we believe these are promising results on a much larger scale.

## 2) Results for 1B model with benchmarks
Beyond just loss values, we have been aiming to include **downstream evaluation benchmarks** -- which are ultimately the main interest -- and have obtained results for a **1B model on long runs of 100B and 460B tokens**.

To be more specific, we used scaling rules by Bi et al. [1] to determine an estimate of the optimal peak learning rate and batch size for two dataset sizes of 100B and 460B tokens of the FineWeb [2] dataset. Notably, the training was stable for both cosine and the constant LR, thus we were able to run the experiments **without issues or tuning, out of the box**.  With the results in the tables below, we find a **clear agreement between the performance of both cosine and the cooldown schedule**, which validates the method beyond just loss values.

## 3) Annealing cosine to 0
In addition, we have added ablations for annealing the cosine schedule to 0, the same final learning rate as for the cooldown schedule. While it helps achieve lower loss values (Figure 2), we find that overall it hurts benchmark metrics (Figure 3 and Tables below). We posit that when annealing to 0, the shape of the cosine curve results in a large portion of training spent with a LR too close to 0; the final LR should therefore also be tuned, which adds complexity to cosine.

All results, including a detailed description of the experimental setup, the evaluation metrics throughout training, and ablations of cosine to 0, will also be available in the updated version of the paper.

---

## Table 1: Results on 100B tokens
* Batch size: 1.86M tokens
* Total steps: 55’000
* Peak Learning Rate: 0.0008

| Schedule              | Aggregated Score | MMLU  | ARC   | OpenBookQA | PIQA  | HellaSwag | CommonSenseQA | SIQA  | Winogrande |
| --------------------- | ---------------- | ----- | ----- | ---------- | ----- | --------- | ------------- | ----- | ---------- |
| Cosine to 10%         | 46.26            | 30.21 | 41.25 | 35.40      | 76.10 | 56.30     | 36.10         | 40.70 | 54.00      |
| Cosine to 0           | 45.88            | 30.70 | 42.35 | 33.80      | 75.50 | 55.60     | 36.20         | 40.20 | 52.70      |
| 1-Sqrt Cooldown (20%) | 46.23            | 30.84 | 41.80 | 36.00      | 76.50 | 55.80     | 37.00         | 39.60 | 52.30      |
| Linear Cooldown (20%) | 46.20            | 30.71 | 41.25 | 35.80      | 77.00 | 55.90     | 36.70         | 39.50 | 52.70      |

## Table 2: Results on 460B tokens
* Batch size: 2.1M tokens
* Total steps: 220’000
* Peak Learning Rate: 0.0008

| Schedule            | Aggregated Score | MMLU  | ARC   | OpenBookQA | PIQA  | HellaSwag | CommonSenseQA | SIQA  | Winogrande |
| ------------------- | ---------------- | ----- | ----- | ---------- | ----- | --------- | ------------- | ----- | ---------- |
| Cosine to 0         | 48.03            | 31.25 | 45.20 | 37.60      | 78.10 | 59.90     | 37.70         | 39.90 | 54.60      |
| 1-Sqrt Cooldown 5%  | 47.91            | 31.71 | 44.10 | 37.80      | 77.40 | 59.60     | 37.30         | 39.50 | 55.90      |
| Linear Cooldown 5%   | 47.84            | 31.65 | 44.05 | 38.00      | 77.40 | 59.50     | 36.70         | 39.40 | 56.00      |
| Linear Cooldown 10% | 47.98            | 31.78 | 44.15 | 38.20      | 77.30 | 60.00     | 36.90         | 39.50 | 56.00      |
| Linear Cooldown 20% | 47.92            | 31.84 | 45.05 | 37.20      | 77.80 | 59.20     | 36.90         | 39.70 | 55.70      |


---

## References
[1] Bi, X., Chen, D., Chen, G., Chen, S., Dai, D., Deng, C., Ding, H., Dong, K., Du, Q., Fu, Z., et al. Deepseek LLM: Scaling Open-Source Language Models with Longtermism. arXiv preprint arXiv:2401.02954, 2024.

[2] Penedo, G., Kydlíček, H., Allal, L. B., Lozhkov, A., Mitchell, M., Raffel, C., Werra, L. V., and Wolf, T. The fineweb datasets: Decanting the web for the finest text data at scale, 2024. URL https://arxiv.org/abs/2406.17557.

[3]  Lozhkov, A., Allal, L. B., Werra, L. V., and Wolf, T. FineWeb-Edu, 2024. URL https://huggingface.co/datasets/HuggingFaceFW/fineweb-edu.

---

### Decision · Program_Chairs · 2024-09-25

**Decision:**

Accept (spotlight)

**Comment:**

The paper offers a practical and well-validated approach to optimizing training schedules for LLMs, with significant implications for continual learning and reducing computational costs in scaling law studies. While the technical novelty is modest, the thorough experimental analysis and potential impact make this work a strong contribution to the field. All reviewers are in favor of this paper. This is a clear accept.